# SARS-CoV and SARS-CoV-2 are transmitted through the air between ferrets over more than one meter distance

Jasmin S. Kutter[1], Dennis de Meulder[1], Theo M. Bestebroer[1], Pascal Lexmond[1], Ard Mulders[1], Mathilde Richard [1], Ron A. M. Fouchier [1] & Sander Herfst [1✉]

SARS-CoV-2 emerged in late 2019 and caused a pandemic, whereas the closely related SARS-CoV was contained rapidly in 2003. Here, an experimental set-up is used to study transmission of SARS-CoV and SARS-CoV-2 through the air between ferrets over more than a meter distance. Both viruses cause a robust productive respiratory tract infection resulting in transmission of SARS-CoV-2 to two of four indirect recipient ferrets and SARS-CoV to all four. A control pandemic A/H1N1 influenza virus also transmits efficiently. Serological assays confirm all virus transmission events. Although the experiments do not discriminate between transmission via small aerosols, large droplets and fomites, these results demonstrate that SARS-CoV and SARS-CoV-2 can remain infectious while traveling through the air. Efficient virus transmission between ferrets is in agreement with frequent SARS-CoV-2 outbreaks in mink farms. Although the evidence for virus transmission via the air between humans under natural conditions is absent or weak for SARS-CoV and SARS-CoV-2, ferrets may represent a sensitive model to study interventions aimed at preventing virus transmission.

[1] Department of Viroscience, Erasmus University Medical Center, Rotterdam, The Netherlands. ✉email: s.herfst@erasmusmc.nl

In December 2019, pneumonia cases were reported in China, caused by a virus that was closely related to the severe acute respiratory syndrome coronavirus (SARS-CoV)[1,2]. In 2003, the SARS-CoV outbreak affected 26 countries and resulted in more than 8000 human cases of infection of whom almost 800 died[3]. In contrast to SARS-CoV, the new coronavirus, named SARS-CoV-2, spread around the world in only a few months, with over 30 million cases and more than 900.000 deaths by the end of September 2020[4]. So far there is no unambiguous experimental or observational evidence on the main mode of transmission of SARS-CoV-2. However, given that most out-breaks occurred in clusters of people in close contact and in household settings, international health authorities conclude that SARS-CoV-2 is primarily transmitted within a short distance between individuals via direct and indirect contact, or respiratory droplets with little support for an important contribution of transmission via the air[5]. To prevent transmission via both routes, the World Health Organization and governments have advised control measures such as frequent hand washing and physical distancing to mitigate the rapid spread of SARS-CoV-2. In addition, in many countries, the use of face masks is encouraged or enforced in public buildings or public transportation where physical distancing is not always possible.

We and others previously used ferret models to show that SARS-CoV can be transmitted via direct contact and that SARS-CoV-2 can be transmitted via the air over 10 cm distance[6–8]. To study if SARS-CoV and SARS-CoV-2 can maintain their infectivity when bridging a distance of more than one meter through the air, an experimental ferret transmission set-up was developed. After validation of the set-up with the A/H1N1 influenza virus, we subsequently demonstrated that both SARS-CoV and SARS-CoV-2 can be transmitted over a one-meter distance via the air.

## Results

**Transmission of A/H1N1 virus between ferrets.** To investigate coronavirus transmission via the air over more than a meter distance, a transmission set-up was built in which individual donor and indirect recipient ferret cages were connected through a hard duct system consisting of horizontal and vertical pipes with multiple 90° turns. The airflow was directed upwards from the donor to the indirect recipient animal and air traveled on average 118 cm through the tube (Fig. 1). A steel grid was placed between each cage and tube opening to prevent spill-over of food, feces, and other large particles.

The transmission set-up was first tested using A/H1N1 influenza virus A/Netherlands/602/2009, which was previously shown to be transmitted efficiently through the air between ferrets over 10 cm distance[9] (Table 1). Four individually housed donor animals were inoculated intranasally with $10^6$ TCID$_{50}$ (median tissue culture infectious dose) of A/H1N1 virus and the next day indirect recipient ferrets were placed in separate cages above those of the donor ferrets. Throat and nasal swabs were collected from the donor and indirect recipient animals on alternating days to prevent cross-contamination, followed by virus detection by qRT-PCR and virus titration. Swabs were collected from the donor and indirect recipient animals until 7 days post-inoculation (dpi) and 13 days post-exposure (dpe), respectively. A/H1N1 virus was detected until 7 dpi in donor animals, with the highest RNA levels until 5 dpi (Fig. 2a). Attempts to isolate infectious virus were successful in all four animals until 5 dpi and in one animal until 7 dpi (Fig. 3a). A/H1N1 virus was transmitted to indirect recipient ferrets in four out of four independent transmission pairs between 1 and 3 dpe onwards, as demonstrated by the presence of viral RNA in throat and nose swabs. Infectious A/H1N1 virus was isolated from three out of four indirect recipient animals with similar peak virus titers

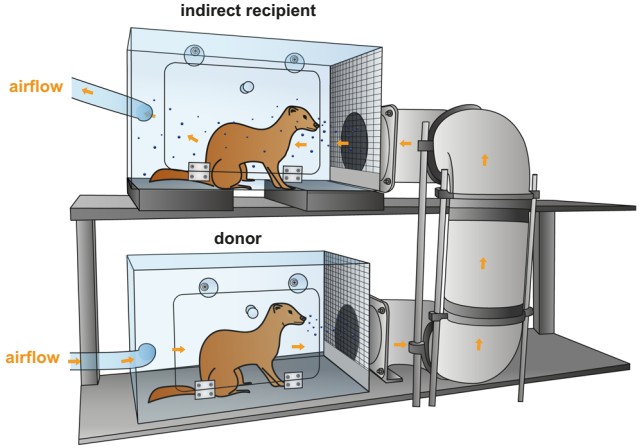

**Fig. 1 Experimental transmission set-up.** Schematic representation of the set-up to assess transmission over >1 m distance. An inoculated donor ferret is housed in the bottom cage and the next day, an indirect recipient ferret is added to the top cage. The cages are connected through a hard duct system consisting of four 90° turns. The system is built of several horizontal and vertical 15 cm wide PVC pipes that allow upward airflow from the donor to the indirect recipient animal. The average length of the duct system is 118 cm with the shortest and longest length 73 and 163 cm, respectively. A steel grid is placed over the inlet and outlet of the duct system. The bottom five cm of the grid was closed to prevent spill-over of food, feces, and other large particles into the tube system. Orange arrows indicate the direction of airflow (100 L/min). Set-ups were placed in class III isolators in a biosafety level 3+ laboratory.

and duration of virus shedding as observed in the donor animals. In these three animals, virus titers ranged from $10^{1.5}$ to $10^{6.0}$ TCID$_{50}$/mL, showing that these indirect recipient ferrets were productively infected (Fig. 3a). Besides nasal discharge, no other signs of illness were observed in the A/H1N1 virus-positive donor and indirect recipient animals (Figs. 2a and 3a). Three of four A/H1N1 virus-positive animals seroconverted 15 dpi/dpe, and the hemagglutination inhibition titers were similar in donor and indirect recipients animals. The indirect recipient animal with low RNA levels and no infectious virus did not seroconvert (Fig. 4a).

**Transmission of SARS-CoV and SARS-CoV-2 between ferrets over a one-meter distance.** Upon validation of the experimental transmission set-up with the A/H1N1 virus, the transmissibility of SARS-CoV and SARS-CoV-2 over more than one-meter distance was assessed, using the same procedures as for the A/H1N1 virus. Four donor animals were inoculated intranasally with either $6 × 10^5$ TCID$_{50}$ of SARS-CoV-2 (isolate BetaCoV/Munich/BavPat1/2020) or $1.6 × 10^6$ TCID$_{50}$ of SARS-CoV (isolate HKU39849). All donor animals were productively infected, as demonstrated by the robust and long-term virus shedding (Fig. 2, Fig. 3). SARS-CoV-2 RNA levels peaked around 3 and 5 dpi and were detected up to 13 dpi in one animal and up to 15 dpi, the last day of sample collection, in the other three animals. In contrast, SARS-CoV RNA levels peaked immediately at 1 dpi. Whereas SARS-CoV-2 inoculated animals did not display any symptoms of the disease, SARS-CoV donor animals became less active and exhibited breathing difficulties from 7 dpi onwards, warranting euthanasia by 9 dpi, when all animals were still SARS-CoV RNA positive in the throat and nasal swabs (Fig. 2).

Interestingly, both SARS-CoV-2 and SARS-CoV transmitted to indirect recipient animals via the air over more than one-meter distance. SARS-CoV-2 was transmitted in two out of four independent transmission pairs at 3 dpe, with peak viral RNA levels at 7 dpe and throat and nasal swabs still positive for viral RNA at

**Table 1 Virus transmission to recipient ferrets over various distances.**

| Virus | Distance between donor and recipient | Recipient ferrets | | | |
| | | Transmission | Onset shedding (dpe) | Peak virus shedding (dpe) | Peak virus titer (log₁₀TCID₅₀/mL) |
|---|---|---|---|---|---|
| A/H1N1 | 10 cm[9] | 4/4 | 3, 3, 1, 3[a] | 3, 3, 5, 5 | 4.8, 5.3, 4.5, 5.0 |
| | >1 m | 4/4 | 5, 1, 3, – [a] | 7, 3, 3, – | 5.3, 5.5, 6.0, – |
| SARS-CoV-2 | DC[6] | 4/4 | 3, 3, 1, 3[b] | 9, 7, 5, 7 | 3.5, 2.9, 2.3, 3.1 |
| | 10 cm[6] | 3/4 | 7, 3, 3[b] | 11, 9, 5 | 4.3, 3.0, 1.7 |
| | >1 m | 2/4 | 1, 3[b] | 7, 5 | 1.6, 3.7 |
| SARS-CoV | DC[7c] | 2/2 | 2, 2[b] | 8, 8 | 4.1[d] |
| | >1 m | 4/4 | 1, 1, 1, 3[b] | 5, 3, 5, 3 | 4.0, 3.6, 3.4, 2.6 |

[a]based on virus titers; [b]based on qRT-PCR Ct-value. DC: direct contact. [c]different transmission set-up and inoculation route (intratracheally); [d]average of two animals; TCID₅₀ equivalent was calculated from a standard curve of serial dilutions of the SARS-CoV virus stock.

**Fig. 2 Virus RNA shedding in ferrets.** A/H1N1 (**a**), SARS-CoV-2 (**b**), and SARS-CoV (**c**) RNA were detected by qRT-PCR in the throat (gray) and nasal (white) swabs collected from a donor (bars) and recipient (circles) ferrets every other day. An individual donor-recipient pair is shown in each panel.

15 dpe, the last day of the experiment (Fig. 2b). Similar to the donor animals, the indirect recipient ferrets did not show any signs of illness. SARS-CoV was transmitted to four out of four indirect recipient ferrets on 1 or 3 dpe, with peak viral RNA levels at 3 to 5 dpe (Fig. 2c). Similar to the donor animals, indirect recipient animals exhibited breathing difficulties and became less active and were consequently euthanized for ethical reasons at 11 dpe, at which time the throat and nasal swabs were still positive for SARS-CoV RNA.

All SARS-CoV and SARS-CoV-2 positive indirect recipient ferrets had seroconverted at 11 and 17 dpe, respectively (Fig. 4). The two indirect recipient ferrets, in which no SARS-CoV-2 was detected, did not seroconvert. Despite the different inoculation routes and doses of the donors that were given a high virus dose in a large volume of liquid and indirect recipient animals that

likely received a lower infectious dose via the air, the kinetics of virus shedding was similar in all animals, both in terms of duration and virus RNA levels. This indicated a robust replication of both SARS-CoV-2 and SARS-CoV upon transmission via the air, independent of the infectious dose and route. In general, SARS-CoV and SARS-CoV-2 RNA levels were higher in the throat swabs as compared to the nasal swabs. From each SARS-CoV and SARS-CoV-2 RNA positive animal, infectious virus was isolated in VeroE6 cells from the throat and nasal swabs for at least two consecutive days (Fig. 3).

**Investigating the potential of fomite transmission via the fur of ferrets.** In SARS-CoV-2 outbreaks on mink farms in the

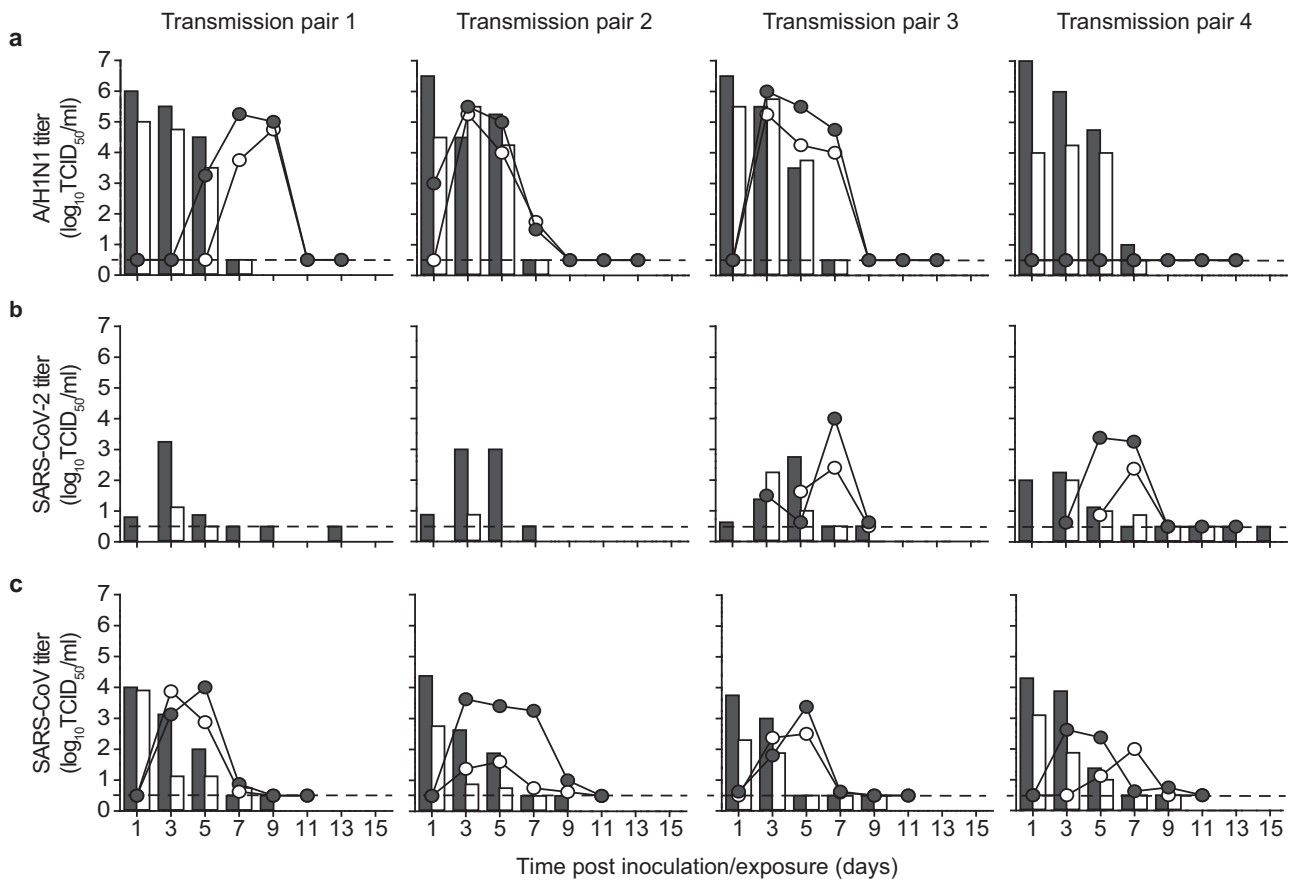

**Fig. 3 Infectious virus shedding in ferrets.** A/H1N1 virus (**a**), SARS-CoV-2 (**b**), and SARS-CoV (**c**) titers were detected in the throat (gray) and nasal (white) swabs collected from inoculated donor (bars) and indirect recipient (circles) ferrets. An individual donor-recipient pair is shown in each panel. The dotted line indicates the detection limit.

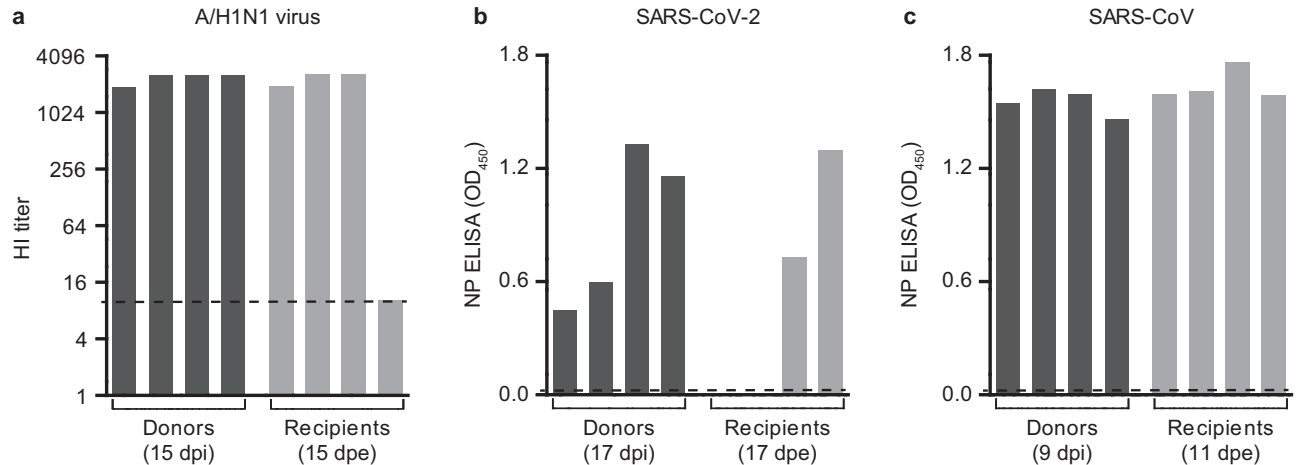

**Fig. 4 Antibody responses in donor and recipient ferrets.** Sera were collected from donor and recipient ferrets on the indicated days. Antibody responses against A/H1N1 virus (**a**) were measured by hemagglutination inhibition (HI) assay, whereas responses against SARS-CoV-2 (**b**) and SARS-CoV (**c**) were assessed using a nucleoprotein (NP) ELISA. Dotted lines indicate the detection limit of each assay. OD: Optic density.

Netherlands, a potential route of virus transmission through aerosolized fomites originating from bedding, fur, and food has been suggested[10]. Although the current transmission set-up was designed to prevent spill-over of large pieces like food and feces from donor to recipient cages, smaller particles such as aerosolized fur or dust from the carpet tiles in the cages, could potentially still be transmitted to the recipient cage. This has very recently been demonstrated in the guinea pig model where a virus-immune animal, whose body was contaminated with influenza virus, transmitted the virus through the air to an indirect recipient animal[11]. Indeed, measurements with an aerodynamic particle sizer in our set-up showed that particles >10 μm were present in the outlet of donor cages, but also at the inlet of the recipient cages, suggesting that despite the distance between the cages, larger particles were carried to the recipient animals due to the high flow rate (Fig S1). To study if fur could serve as a

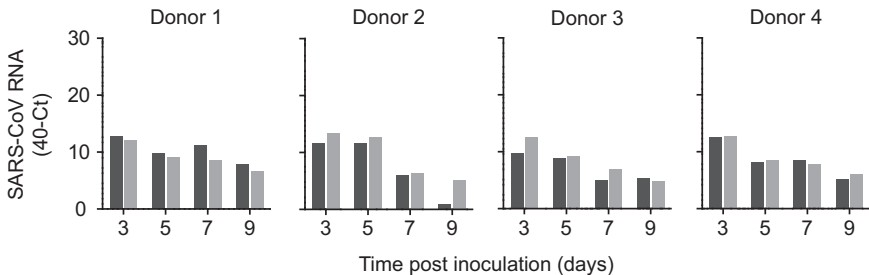

**Fig. 5 Detection of SARS-CoV RNA on the fur of donor ferrets.** SARS-CoV RNA was detected by qRT-PCR in swabs collected from the fur on the left (dark gray) and right (light gray) flank of all four donor ferrets. An infectious virus was not detected in these samples.

carrier for infectious virus, fur swabs from the left and right flank of SARS-CoV inoculated donor ferrets were also collected in the last experiment from 3 to 9 dpi. SARS-CoV RNA was detected in fur swabs of all donor ferrets indicating that the fur of ferrets was contaminated with the virus and therefore can be a potential source for aerosolized fomite transmission (Fig. 5). Given the observed grooming behavior of ferrets and the high viral loads detected in the URT, grooming was most likely the dominant route of virus transfer to the fur. SARS-CoV RNA levels were on average 240-fold (7,9 Ct) lower than those in the throat and nasal swabs of the same donor ferrets. Importantly, no infectious virus was isolated from these fur samples. The inability to detect infectious virus in fur samples was in agreement with the inability to detect infectious virus in respiratory samples with similarly low viral RNA levels.

**Sequence analysis of viruses isolated from ferrets**. To study if genetic changes might have contributed to the transmission of SARS-CoV-2 via the air between ferrets, Illumina next-generation sequencing was performed on the virus inoculum and on throat swabs of all four donors (3 dpi) and two indirect recipient ferrets (5 and 7 dpe). Single nucleotide polymorphisms (SNP) which were present in >5% of the total number of reads were called (Table S1A). Mainly, two substitutions were detected in the sequence of all virus isolates: N501T and S686G. Both residues are located in the spike protein and have been detected previously in ferrets[6]. N501T and S686G were present in all donor ferrets in the majority of reads (53–99%) and in both indirect recipient ferrets in >99% of reads. The SNP analysis of the virus isolate used to inoculate ferrets (passage 3 of the virus stock) revealed that only the substitution S686G was present in >5% of the reads (Table S1B). Given that these substitutions were already present in the donor ferrets, it is likely that they were selected due to adaptation to the new host, rather than having substantial importance for transmission. A replacement of asparagine by threonine at position 501 decreases the binding affinity of the spike protein to human ACE2 but might favor binding to ferret ACE2[12]. An additional L1035F substitution was detected in Nsp3 in the throat swab of a donor ferret and a synonymous mutation C10757T in Nsp5 in the throat swab of an indirect recipient ferret.

## Discussion

Here, it is shown that SARS-CoV can be transmitted through the air between ferrets and that both SARS-CoV and SARS-CoV-2 are transmissible through the air between ferrets over more than a meter distance, similar to a control A/H1N1 influenza virus.

In the transmission set-up described here, ferret cages were connected by a hard duct system with four 90° turns and a flow rate of approximately 100 L/min. The shortest and longest distance between the inlet and outlet of the duct system was 73 and

163 cm, respectively, so that viruses shed by the donor animal had to bridge an average distance of 118 cm before reaching the cage of the indirect recipient ferret. Based on airflow fundamentals, it is anticipated that the minimal distance of the path followed by the particles through the duct is 1 m. The duct system was designed to have an upward airflow, with the aim to prevent large particles to reach the outlet of the duct system. Unfortunately, particles >10 μm that originated from the donor cage were detected in the indirect recipient cage, which was likely due to the relatively high flow rate. As a consequence, the set-up described here does not allow the discrimination between transmission of viruses via aerosols, droplets, and aerosolized fomites, and therefore transmission between ferrets can occur via either route.

Ferrets and minks both belong to the *Mustelinea* subfamily of the *Mustelidae* family. Minks are the first animal species for which SARS-CoV-2 outbreaks have been reported, and to date, outbreaks have been detected on 53 mink farms in the Netherlands and on several mink farms in Denmark, Spain, and the USA[10,13]. In investigations of the first two outbreaks, 119 out of 120 serum samples collected from minks were positive, indicating that SARS-CoV-2 had spread readily through the population[10]. The high infection rate among minks together with the productive SARS-CoV-2 infection in ferrets suggests that mustelids are highly susceptible to infection with SARS-CoV-2, perhaps even more so than humans.

Epidemiological studies in humans in 2003 demonstrated that SARS-CoV transmission occurred often during the second week of illness. Virus excretion in respiratory secretions and stool followed a Gaussian distribution and peaked approximately 10 days after symptom onset when patients were often already hospitalized[14–17]. Hence, most cases of SARS-CoV human-to-human transmission occurred in healthcare settings, predominantly when adequate infection control precautions were absent. Virus transmission via the air was limited to hospital procedures where mechanical aerosol formation could not be prevented. The fact that SARS-CoV was transmitted efficiently via the air between ferrets thus does not align well with the lack of evidence for efficient SARS-CoV virus transmission via the air between humans under natural conditions. In the four indirect recipient animals that became infected with SARS-CoV upon transmission via the air, virus replication peaked as early as 3 to 5 dpe (Fig. 3). This demonstrated that SARS-CoV replicates remarkably faster to peak titers in ferrets as compared to the 10 days after symptom onset in humans, and indicated that ferrets are also highly susceptible for SARS-CoV as observed for SARS-CoV-2, which may have contributed to the observed high efficiency of transmission in the ferret model.

Distinctive from what was described for SARS-CoV, infection with SARS-CoV-2 is characterized by long-term shedding of virus RNA in patients, characterized by peak RNA levels on the day of symptom onset or earlier and infectious virus has primarily been successfully isolated in the initial phase of illness[17–20]. During several

outbreaks in churches, nursing homes, call centers, cruise ships, and restaurants a potential role for SARS-CoV-2 transmission via the air has been debated but remained inconclusive as other transmission routes could not be excluded[21–25]. In a few studies, low concentrations of SARS-CoV-2 RNA were detected in air samples collected in healthcare settings[26–29]. However, in only one study infectious SARS-CoV-2 was isolated from air samples collected in a hospital room, 2–4.8 m away from patients[30]. Despite the lack of evidence that exposure to SARS-CoV-2 over substantial distances poses a high infection risk, the debate about the potential role of small aerosols and large droplets in SARS-CoV-2 transmission through the air remains.

It was recently shown for influenza virus in the guinea pig model that virus transmission through the air is also possible via aerosolized fomites originating from fur; animals transmitted the virus to 25% of the indirect recipient animals when $10^8$ PFU of influenza virus was applied on fur, compared to 88% via airways and fur upon intranasal inoculation[11]. In this study, SARS-CoV RNA was detected on fur swabs from four out of four donor animals but no infectious virus was isolated. In contrast, in the guinea pig study of Asadi et al., up to 650 PFU of infectious influenza virus was recovered from the fur of intranasally inoculated animals, which is not a surprise since influenza viruses replicate to much higher titers than SARS-CoV-2 as also shown in Fig. 3. Thus, although transmission via aerosolized fomite particles cannot be excluded in this study, the low amounts of viral RNA and undetectable levels of infectious virus in fur as compared to those in the guinea pig studies make this a less likely route here.

The efficiency of transmission via the air depends on the anatomical site of virus excretion, the amount and duration of infectious virus shedding in the air, the ability of the virus to remain infectious in the air, and the infectious dose required to initiate an infection in an individual. It was recently shown that influenza A viruses are transmitted via the air from the nasal respiratory epithelium of ferrets[31]. In this study, SARS-CoV and SARS-CoV-2 RNA were detected in nose and throat swabs of all infected ferrets. In COVID-19 patients, SARS-CoV-2 RNA was also easily detected in upper respiratory tract (URT) specimens, however, the detection rate of SARS-CoV RNA in URT specimens of SARS patients was low, with SARS-CoV RNA detection by RT-PCR in only 32% to 68% of the tested patients[14,18,32,33]. This lower detection rate, likely as a result of lower or no replication of SARS-CoV in the upper respiratory tract, may explain why SARS-CoV was less efficiently transmitted between humans than SARS-CoV-2.

The RNA levels and infectious SARS-CoV-2 titers detected in respiratory swabs collected from ferrets and humans were similar[34]. However, the duration and moment of peak virus shedding are different, as described above. The susceptibility to infection is probably different between ferrets and humans, especially given the difference in the efficiency of spread observed in ferrets and minks on one hand and humans on the other hand. With respect to the ferret model it should be noted that in the experimental set-up with uni-directional airflow described here, indirect recipient animals are constantly at the right place at the right moment, which may contribute to the relatively high efficiency of virus transmission via the air. It is also important to note that superspreading events played a critical role in the epidemiology of SARS-CoV and SARS-CoV-2. Several superspreading events were identified during the SARS-CoV outbreak and there is growing evidence for such events during the COVID-19 pandemic[35–38]. However, it is still unknown which transmission route is predominantly involved in these events[39].

Altogether, our data on the transmissibility of SARS-CoV and SARS-CoV-2 demonstrate qualitatively that SARS-CoV and SARS-CoV-2 can remain infectious when transmitted through the air over more than one-meter distance. However, quantitatively, the data should be interpreted with caution and no conclusions can be drawn about the importance of transmission via the air in the spread of SARS-CoV-2 in the human population. Although the evidence for virus transmission via the air between humans under natural conditions is absent or very weak for both SARS-CoV and SARS-CoV-2, ferrets may represent a sensitive model to study intervention strategies aimed at preventing virus transmission.

## Methods

**Viruses and cells.** Influenza A/H1N1 virus (isolate A/Netherlands/602/2009) was passaged once in embryonated chicken eggs followed by two passages in Madin-Darby Canine Kidney (MDCK) cells (ATCC) in Eagle's minimal essential medium (EMEM; Lonza) supplemented with 100 IU mL$^{-1}$ penicillin-100 µg mL$^{-1}$ streptomycin mixture (Lonza), 2 mM L-glutamine (Lonza), 1.5 mg mL$^{-1}$ sodium bicarbonate (Lonza), 10 mM Hepes (Lonza), 1× nonessential amino acids (Lonza) and 20 µg mL$^{-1}$ trypsin (Lonza). MDCK cells were inoculated at an moi of 0.01. The supernatant was harvested at 72 hpi, cleared by centrifugation, and stored at −80 °C. MDCK cells were maintained in EMEM supplemented with 10% fetal bovine serum (Greiner), 100 IU mL$^{-1}$ penicillin-100 µg mL$^{-1}$ streptomycin mixture (Lonza), 200 mM L-glutamine (Lonza), 1.5 mg mL$^{-1}$ sodium bicarbonate (Lonza), 10 mM Hepes (Lonza), and 1× nonessential amino acids (Lonza).

SARS-CoV-2 (isolate BetaCoV/Munich/BavPat1/2020; kindly provided by Prof. Dr. C. Drosten) and SARS-CoV (isolate HKU39849, kindly provided by Prof. Dr. M. Peiris) were propagated to passage 3 and 9, respectively, in Vero E6 cells (ATCC) in Opti-MEM (1×) + GlutaMAX (Gibco), supplemented with penicillin (10,000 IU mL$^{-1}$, Lonza) and streptomycin (10,000 IU mL$^{-1}$, Lonza). Vero E6 cells were inoculated at an moi of 0.01. The supernatant was harvested at 72 hpi, cleared by centrifugation, and stored at −80 °C. Vero E6 cells were maintained in Dulbecco's Modified Eagle Medium (DMEM, Gibco or Lonza) supplemented with 10% fetal bovine serum (Greiner), 100 IU mL$^{-1}$ penicillin-100 µg mL-1 streptomycin mixture (Lonza), 2 mM L-glutamine (Lonza), 1.5 mg mL$^{-1}$ sodium bicarbonate (Lonza) and 10 mM Hepes (Lonza). Both cell lines were maintained at 37 °C and 5% $CO_2$.

**Ferret transmission experiment.** Animals were housed and experiments were performed in strict compliance with the Dutch legislation for the protection of animals used for scientific purposes (2014, implementing EU Directive 2010/63). Influenza virus, SARS-CoV-2, and Aleutian Disease Virus seronegative 6 month-old female ferrets (Mustela putorius furo), weighing 640–1215 g, were obtained from a commercial breeder (TripleF, USA). The research was conducted under a project license from the Dutch competent authority (license number 248 AVD1010020174312) and the study protocols were approved by the institutional Animal Welfare Body (Erasmus MC permit number 17-4312-03, 17-4312-05, and 17-4312-06). Animal welfare was monitored on a daily basis. Humane endpoint criteria were defined as follows: animal does not eat or drink anymore (1), >20% loss of body weight (2), moderate to serious circulation problems or breathing difficulties (3), moderate to serious behavioral and motor changes (4) and display of moderate to serious clinical symptoms (5).

Virus inoculation of ferrets was performed under anesthesia with a mixture of ketamine/medetomidine (10 and 0.05 mg kg$^{-1}$, respectively) antagonized by atipamezole (0.25 mg kg$^{-1}$). Swabs were taken under light anesthesia using ketamine to minimize animal discomfort. Four donor ferrets were inoculated intranasally with $10^6$ TCID$_{50}$ of A/H1N1 virus, $6 \times 10^5$ TCID$_{50}$ of SARS-CoV-2, or $1.6 \times 10^6$ TCID$_{50}$ of SARS-CoV (250 µL instilled dropwise in each nostril) and were housed individually in a cage. One day later, indirect recipient ferrets were added to a cage placed above the donor cage. Both cages were connected by a 15 cm wide duct system with four 90° turns. The average length of the duct system was 118 cm long, with an upward airflow of 100 L min$^{-1}$ from the donor to the indirect recipient cage (Fig. 1). Throat and nasal swabs were collected using dry swabs (Copan, cat. 155CS01) from the ferrets every other alternating day to prevent cross-contamination. For the assessment of A/H1N1 virus transmission between ferrets, swabs of donor and indirect recipient animals were collected until 7 dpi and 13 dpe, respectively. Swabs of donor and indirect recipient animals for the SARS-CoV-2 experiment were collected until 15 dpi/dpe. Swabs of SARS-CoV inoculated donor animals were collected until 9 dpi and of indirect recipient animals until 11 dpe. For SARS-CoV inoculated animals, fur samples were collected from 3 dpi onwards by swabbing the left and right flank of animals with swabs (Copan, cat. 155CS01) wetted in virus transport medium (VTM) consisting of Minimum Essential Medium (MEM)–Eagle with Hank's BSS and 25 mM Hepes (Lonza), glycerol 99% (Sigma Aldrich), lactalbumin hydrosylate (Sigma Aldrich), 10 MU polymyxin B sulfate (Sigma Aldrich), 5 MU nystatin (Sigma Aldrich), 50 mg/mL gentamicin (Gibco) and 100 IU mL$^{-1}$ penicillin 100 µg mL$^{-1}$ streptomycin mixture (Lonza). All swabs were stored at −80 °C in VTM for end-point titration in Vero E6 cells as described below. Ferrets were euthanized by heart puncture under anesthesia. Blood was collected in serum-separating tubes (Greiner) and

processed according to the manufacturer's instructions. Sera were heated for 30 min at 60 °C and used for the detection of virus-specific antibodies as described below. All animal experiments were performed in class III isolators in a negatively pressurized ABSL3+ facility. Average temperature and relative humidity in the isolators were 22.9 °C (±0.2 °C) and 53.2% (±9.0%), respectively (Fig S2). After each experiment, transmission set-ups were disassembled in the class III isolators and subsequently decontaminated by two rounds of formaldehyde fumigation. After the second fumigation, the formaldehyde gas was neutralized with ammonia. The effectiveness of the formaldehyde fumigation was validated each round with biological indicator strips (EZTest, MesaLabs). After successful fumigation, isolators and transmission set-ups were thoroughly cleaned with soap and water.

**RNA isolation and qRT-PCR**. Virus RNA was isolated from swabs using an in-house developed high-throughput method in a 96-well format. Sixty microliters of the sample were added to 90 μL of MagNA Pure 96 External Lysis Buffer. A known concentration of phocine distemper virus (PDV) was added to the sample as an internal control for the RNA extraction[40]. The 150 μL of sample/lysis buffer was added to a well of a 96-well plate containing 50 μL of magnetic beads (AMPure XP, Beckman Coulter). After thorough mixing by pipetting up and down at least 10 times, the plate was incubated for 15 min at room temperature. The plate was then placed on a magnetic block (DynaMag™–96 Side Skirted Magnet, ThermoFisher Scientific) and incubated for 3 min to allow the displacement of the beads towards the side of the magnet. Supernatants were carefully removed without touching the beads and beads were washed three times for 30 s (sec) at room temperature with 200 μL/well of 70% ethanol. After the last wash, a 10 μL multi-channel pipet was used to remove residual ethanol. Plates were air-dried for 6 min at room temperature. Plates were removed from the magnetic block and 50 μL of elution buffer (Roche) was added to each well and mixed by pipetting up and down 10 times. Plates were incubated for 5 min at room temperature and then placed back on the magnetic block for 2 min to allow separation of the beads. Supernatants were pipetted in a new plate and RNA was kept at 4 °C. Eight microliters of RNA were directly pipetted into a mix for qRT-PCR, containing 0.4 μL of primers and probe mix targeting the M gene of A/H1N1 virus[41], the E gene of SARS-CoV-2[42], or the NP gene of SARS-CoV[43], 0.4 μL of primers and probe mix targeting the HA gene of PDV[41], 4 μL of TaqMan™ Fast Virus 1-Step Master Mix (ThermoFisher Scientific) and 6.2 μL of PCR grade water (for primer and probe sequences see Table S2). Amplification and detection were performed on an ABI7700 (Thermo Fisher Scientific) using the following program: 5 min 50 °C, 20" 95 °C, [3" 95 °C, 31" 58 °C] × 45 cycles.

**Virus titrations**. Throat and nasal swabs were titrated in quadruplicates in either MDCK or VeroE6 cells. Briefly, confluent cells were inoculated with 10-fold (A/H1N1 virus) and 3-fold (SARS-CoV-2 and SARS-CoV) serial dilutions of the sample in serum-free EMEM supplemented with 20 μg mL$^{-1}$ trypsin (Lonza) for MDCK cells, or Opti-MEM I (1×) + GlutaMAX, supplemented with penicillin (10,000 IU mL$^{-1}$), streptomycin (10,000 IU mL$^{-1}$), primocin™ (50 mg mL$^{-1}$, Invivogen) for Vero E6 cells. At one hpi, the first three dilutions were washed twice with media, and 200 μL fresh media was subsequently added to the whole plate. For swabs of ferrets from the A/H1N1 virus experiment, supernatants of cell cultures were tested for agglutination activity using turkey erythrocytes three days after inoculation. For swabs of ferrets from the SARS-CoV and SARS-CoV-2 experiments, virus positivity was assessed by reading out cytopathic effects in the cell cultures. Infectious virus titers (TCID$_{50}$ mL$^{-1}$) were calculated from four replicates of each throat and nasal swab using the Spearman-Karber method.

**Serology**. Sera of ferrets from the A/H1N1 virus experiment were tested for virus-specific antibodies using the hemagglutination inhibition (HI) assay[44]. Briefly, ferret antisera were treated with receptor destroying enzyme (Vibrio cholerae neuraminidase) and incubated at 37 °C overnight, followed by inactivation of the enzyme at 56 °C for 1 h. Twofold serial dilutions of the antisera, starting at a 1:10 dilution, were mixed with 25 μL phosphate-buffered saline (PBS) containing four hemagglutinating units of virus and were incubated at 37 °C for 30 min. Subsequently, 25 μL 1% turkey erythrocytes were added, and the mixture was incubated at 4 °C for 1 h. HI titers were read and expressed as the reciprocal value of the highest dilution of the serum that completely inhibited agglutination of virus and erythrocytes. Sera of ferrets from the SARS-CoV and SARS-CoV-2 experiments were tested for virus-specific antibodies using a receptor-binding domain (RBD) enzyme-linked immunosorbent assay (ELISA) as described previously, with some modifications[45]. Briefly, ELISA plates were coated overnight at 4 °C with 100 ng/well with SARS-CoV NP protein (Sino Biological Inc.). After blocking with BlockerTM BLOTTO in TBS (Life technologies) + 0.01% of Tween-20 (Sigma-Aldrich), heat-inactivated sera (diluted 1:100) were added and incubated for 1 h at 37 °C. Bound antibodies were detected using horseradish peroxidase (HRP)-labeled goat anti-ferret IgG (1:10,000; ab112770, Abcam) and 3,3',5,5'-Tetra-methylbenzidine (TMB, Life Technologies) as a substrate. The absorbance of each sample was measured at 450 nm. OD-values were higher than two times the background value of negative serum (0.02) were considered positive.

**Next-generation sequencing**. Amplicons were generated by a SARS-CoV-2 specific multiplex PCR[46] (for primer sequences, see Table S3). Amplicons were purified with 0.8x AMPure XP beads (Beckman Coulter) and 100 ng of DNA was converted into paired-end Illumina sequencing libraries using the KAPA Hyper-Plus library preparation kit (Roche), following the manufacturer's recommendations, to enable subsequent sequencing of multiple libraries in a single Illumina V3 MiSeq flowcell (2 × 300 cycles). Multiplex Adaptors (KAPA Unique Dual-Indexed Adapters Kit (Roche)) with indexes were used. FASTQ files were then imported to the CLC Genomics Workbench v20.0.3 (QIAGEN) for analysis. First, sequences were trimmed off 33 base pairs on both the 3′ and 5′ ends to remove primer sequences and also using Phred quality score threshold of 20. The trimmed sequences were mapped to the reference sequence (GISAID ID EPI_ISL 406862) with the following default parameters (match score = 1, mismatch cost = 2, insertion cost = 3, length fraction = 0.5, and similarity fraction = 8). Variants were called with the Basic Variant Detection tool. Single nucleotide polymorphisms that were present in both the forward and reverse reads with a 100x minimum coverage and a minimum variant count of 5 (5%) were called.

**Particle size measurements**. To determine the number and particle size distribution of droplets and aerosols entering and exiting the tubing system, particles at the outlet of the donor and the inlet of the indirect recipient cage were measured with an aero-dynamic particle sizer (APS; Model Solair 1100+, Lighthouse Worldwide Solutions Benelux BV, Fig S1). Measurements were recorded every minute for a total of 60 min. To determine the number and size of particles produced by the caging environment, as well as by a ferret, measurements were performed with and without an uninfected ferret present in the bottom cage. In addition, the activity of the ferret was observed visually and recorded in 5-min intervals for 60 min.

**Reporting summary**. Further information on research design is available in the Nature Research Reporting Summary linked to this article.

## Data availability

All data are available from the corresponding author (S.H.) on reasonable request. The sequencing raw data were deposited in the NCBI Sequence Read Archive (SRA) under the BioProject PRJNA700531. Source data are provided with this paper.

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

## Acknowledgements

We thank Prof. Dr. Christian Drosten (Charité–Universitätsmedizin Berlin) for providing the SARS-CoV-2 isolate used in this study. This work was financed through an NWO VIDI grant (contract number 91715372), NIH/NIAID contract HHSN272201400008C, and European Union's *Horizon 2020* research and innovation program *VetBioNet* (grant agreement no. *731014*).

## Author contributions

J.K. and S.H. conceived, designed, analyzed, and performed the work and wrote the manuscript. A.M. helped with the design of the transmission set-up. D.M., T.B., P.L., and M.R. helped with performing the work. R.A.M.F. helped with the design of the work, interpretation of the data, and manuscript revision. All authors read and approved the final manuscript.

## Competing interests

The authors declare no competing interests.
