## [Peer Review File · Nature Communications]

Reviewers' Comments:

Reviewer #1:

Remarks to the Author:

Herfst and colleagues report on the airborne-transmission of SARS-CoV and SARS-CoV-2 in the ferret model. The authors revised their experimental approach to assess airborne transmission of virus over a distance of one meter. The non-linear system contained four 90 degree turns that altered airflow. The authors included an influenza A virus as a comparative reference for the transmission model. Although the experimental approach is not innovative, the authors demonstrate airborne transmission of coronaviruses in their ferret model. However, the authors should repeat the SARS-CoV-2 transmission study with the directly infected ferret placed in the upper cage and the naive contact in the lower cage to exclude potential cross-contamination during husbandry practices. The current approach does not address the minimum infectious dose of the coronaviruses for ferrets, which would govern transmission of viruses. The authors should examine the genetic evolution of coronaviruses in this animal model to determine if genetic changes contribute to airborne transmission.

Reviewer #2:

Remarks to the Author:

Recent studies indicate that ferrets infected with SARS-CoV-2 virus do not present with severe respiratory disease, but are able to transmit the virus to naïve animals in both close contact and short-range airborne settings. In this study, Kutter and colleagues employ a modified caging design to expand the distance typically employed between infected and naïve animals, to demonstrate that SARS-CoV-2 (in addition to SARS-CoV-1 and H1N1 influenza viruses) is transmissible at distances over one meter. There is a need to evaluate the capacity for the novel coronavirus to transmit over long distances, and studies such as this can provide important evidence to support public health countermeasures. The manuscript clearly states this rationale, and the use of three viruses in this setup improves contextualizing findings with SARS-CoV-2. That said, while the results are noteworthy, the study is largely descriptive, and additional investigation of the airborne particles responsible for these transmission events would greatly improve the rigor, reproducibility, and utility of this article.

Major Comments:

1. It appears from the text that an aerodynamic particle sizer (APS, line 132-5 and lines 158-9) was employed to measure aerosolized particles in both donor and recipient animal cages. However, this methodology does not appear in the methods, and the authors do not expand upon this information further. Did the authors quantify the relative amount and size range of aerosolized particles present in the donor and/or recipient cages, and did these amounts vary between viruses tested? On what days post-inoculation were these particles measured? This information would greatly contextualize the ferret transmission results presented in this manuscript.
2. While the focus of the manuscript is regarding transmissibility, the divergent virulence between the three viruses tested is still important information to the field. It would be beneficial to see this data in a separate table (or incorporated into Table 1) and not just mentioned incidentally in the text, inclusive of weight loss, temperature, nasal discharge, sneezing incidence, etc between the H1N1 influenza virus, SARS-CoV-1, and SARS-CoV-2 viruses, for both donor and contact ferrets. In this vein, humane endpoint criteria that were employed for SARS-CoV-1 ferrets appear to be missing from the methods and should be added.
3. In the last section of the study, the authors consider fomite transmission, but this section seems underdeveloped and hard to contextualize with the rest of the study. Authors show in Figure 5 that viral RNA from SARS-CoV-1 virus can be recovered from the fur of inoculated ferrets, but this represents only one potential source of virus-containing fomites in the caging environment. Were fur

samples collected from contact ferrets before/after onset of virus replication to support this conclusion? Were other areas of the caging environment (surrounding walls, carpet tiles, etc) similarly sampled? If the inoculated ferret was removed from the contaminated cage prior to the contact ferret being exposed to airflow, would the environmental fomites present in the 'dirty' cage be sufficient to lead to a transmission event? Without inclusion of additional experiments, it would be beneficial if the authors limit their interpretation of this data to detection (or lack thereof) of live virus on fur, and minimize extrapolation of their findings to the larger question of the role fomite transmission may play.

Minor Comments:

1. Lines 115-6, the similar clinical and virological parameters between donor and airborne contact ferrets for all viruses tested is interesting, especially considering the lower dose of virus likely initiating the contact ferret infection. Are the 50% ferret infectious doses known for the viruses studied here, or can this be contextualized to what has been studied in humans?
2. Lines 138-9, the authors conclude that detection of SARS-CoV-2 virus from the fur of directly inoculated ferrets is attributable to grooming. Could this not also be attributed to aerosolized virus released from the inoculated ferret landing on the fur?
3. Authors should specify the temperature and relative humidity at which these experiments were performed in the methods. Authors should also include in the methods section how the modified transmission setup and tubing was decontaminated between experiments.
4. Authors mention the "high flow rate" (line 159) that may have carried particles >10um in diameter upward from the inoculated to the donor cage. Did the authors modulate the flow rate (100 L/min) in any experiments to a lower rate to account for this?

REVIEWER COMMENTS

Reviewer #1 (Remarks to the Author):

Herfst and colleagues report on the airborne-transmission of SARS-CoV and SARS-CoV-2 in the ferret model. The authors revised their experimental approach to assess airborne transmission of virus over a distance of one meter. The non-linear system contained four 90 degree turns that altered airflow. The authors included an influenza A virus as a comparative reference for the transmission model. Although the experimental approach is not innovative, the authors demonstrate airborne transmission of coronaviruses in their ferret model.

1. However, the authors should repeat the SARS-CoV-2 transmission study with the directly infected ferret placed in the upper cage and the naive contact in the lower cage to exclude potential cross-contamination during husbandry practices.

We appreciate the concerns of the reviewer, however we have strict procedures in place ensuring that animals do not get infected due to cross-contamination during husbandry practices. For this specific reason, donor and recipient animals are never handled on the same day (swabs of donor and recipient ferrets are collected on alternating days) and for each animal a separate set of equipment, including gloves and scissors (to cut the swabs) is used. Furthermore, after daily procedures, all equipment is rigorously disinfected with 70% ethanol. In addition, we and others showed in various studies that, despite robust virus replication in the upper respiratory tract of donor ferrets, wildtype avian influenza viruses never transmitted to indirect recipient animals, in contrast to seasonal and pandemic human influenza A viruses¹⁻³. This observation is summarized in supplemental table 3 of one of our previous publications (Linster et al., Cell, 2014). In our opinion, the above mentioned reasons already provide enough evidence that indirect recipient ferrets were not infected due to cross-contamination. Consequently, we consider that it would be unethical to sacrifice additional animals for the sole purpose of another negative control.

2. The current approach does not address the minimum infectious dose of the coronaviruses for ferrets, which would govern transmission of viruses.

This is correct. Investigating the minimum infectious dose of the coronaviruses would require a different experimental set-up and was not the aim of the current study.

3. The authors should examine the genetic evolution of coronaviruses in this animal model to determine if genetic changes contribute to airborne transmission.

We agree with the reviewer. Similar to what was done in our previous SARS-CoV-2 transmission experiments, Illumina next-generation sequencing was performed on the SARS-CoV-2 virus stock and throat samples of SARS-CoV-2 positive donor and indirect recipient ferrets. Two substitutions were consistently detected in the spike protein in samples from all ferrets: the N501T substitution, which is an ACE2 contact residue, and the S686G substitution, which is part of the furin cleavage site. Both substitutions were detected in sequences collected from ferret throat swabs in our previous study as well⁴. Given that these substitutions were already present

in the donor ferrets, it is likely that they were selected due to adaptation to the new host, rather than having a substantial importance for transmission.

The sequencing data were added in a supplementary table (Table S1) and described in the main text. We did not perform next-generation sequencing on samples obtained from ferrets infected with pH1N1 or SARS-CoV, as these were only used as control viruses. The sequencing and sequence analysis was done by Dr. Mathilde Richard, who is now added as a co-author to the manuscript.

Reviewer #2 (Remarks to the Author):

Recent studies indicate that ferrets infected with SARS-CoV-2 virus do not present with severe respiratory disease, but are able to transmit the virus to naïve animals in both close contact and short-range airborne settings. In this study, Kutter and colleagues employ a modified caging design to expand the distance typically employed between infected and naïve animals, to demonstrate that SARS-CoV-2 (in addition to SARS-CoV-1 and H1N1 influenza viruses) is transmissible at distances over one meter. There is a need to evaluate the capacity for the novel coronavirus to transmit over long distances, and studies such as this can provide important evidence to support public health countermeasures. The manuscript clearly states this rationale, and the use of three viruses in this setup improves contextualizing findings with SARS-CoV-2. That said, while the results are noteworthy, the study is largely descriptive, and additional investigation of the airborne particles responsible for these transmission events would greatly improve the rigor, reproducibility, and utility of this article.

Major Comments:

1. It appears from the text that an aerodynamic particle sizer (APS, line 132-5 and lines 158-9) was employed to measure aerosolized particles in both donor and recipient animal cages. However, this methodology does not appear in the methods, and the authors do not expand upon this information further. Did the authors quantify the relative amount and size range of aerosolized particles present in the donor and/or recipient cages, and did these amounts vary between viruses tested? On what days post-inoculation were these particles measured? This information would greatly contextualize the ferret transmission results presented in this manuscript.

We agree with the reviewer and added additional information regarding the particle size measurements in a supplementary figure (Fig S1). It is not possible to record particle sizes during the experiments in the negatively pressurized class 3 isolator, as this would disturb the airflow through the cages and contaminate the particle sizer. We therefore recorded the number and size of droplets and aerosols in a clean transmission set-up with and without a ferret present in the bottom cage. Measurements were performed at the inlet or outlet of the tubing system as illustrated in supplemental figure 1. The measurements at either side were performed separately, because measurements at both ends simultaneously would compete with each other and affect the outcome. In addition, the activity of the ferret was recorded during the

measurements. The number of particles detected without a ferret present in the bottom cage were negligible as compared to the number of particles recorded with a ferret in the cage. Particles > 10 µm were also recorded at the inlet of the upper cage, demonstrating that large droplets were also efficiently transported through the tubing system. Overall, these measurements do not allow us to make any statements about the size of aerosols and droplets that governed virus transmission.

2. While the focus of the manuscript is regarding transmissibility, the divergent virulence between the three viruses tested is still important information to the field. It would be beneficial to see this data in a separate table (or incorporated into Table 1) and not just mentioned incidentally in the text, inclusive of weight loss, temperature, nasal discharge, sneezing incidence, etc between the H1N1 influenza virus, SARS-CoV-1, and SARS-CoV-2 viruses, for both donor and contact ferrets. In this vein, humane endpoint criteria that were employed for SARS-CoV-1 ferrets appear to be missing from the methods and should be added.

We agree with the reviewer that the divergent virulence is interesting, however, the goal of the present experiment was to assess the transmissibility of SARS-CoV-2 between ferrets over more than 1 meter, and not to assess its pathogenicity, because this was previously done by others. Furthermore, daily body weight and temperature measurements would greatly increase the chance of cross-contamination and are thus deliberately omitted in transmission experiments unless warranted by humane endpoint criteria. Overall, clinical signs of illness were as previously described in other studies^{3,5,6}. Humane endpoint criteria were added to the methods section (lines 448-452).

3. In the last section of the study, the authors consider fomite transmission, but this section seems underdeveloped and hard to contextualize with the rest of the study. Authors show in Figure 5 that viral RNA from SARS-CoV-1 virus can be recovered from the fur of inoculated ferrets, but this represents only one potential source of virus-containing fomites in the caging environment. Were fur samples collected from contact ferrets before/after onset of virus replication to support this conclusion? Were other areas of the caging environment (surrounding walls, carpet tiles, etc) similarly sampled? If the inoculated ferret was removed from the contaminated cage prior to the contact ferret being exposed to airflow, would the environmental fomites present in the 'dirty' cage be sufficient to lead to a transmission event? Without inclusion of additional experiments, it would be beneficial if the authors limit their interpretation of this data to detection (or lack thereof) of live virus on fur, and minimize extrapolation of their findings to the larger question of the role fomite transmission may play.

During our last transmission experiment with SARS-CoV, an interesting study by Asadi et al. was published showing that influenza A viruses can also be transmitted - although less efficiently - through the air via aerosolized fur particles⁷. To investigate if this could also be a potential transmission route in our experiments, we thus were only able to collect samples of the fur of donor animals during the SARS-CoV experiment. We did not take swabs from the fur of recipient animals or of any other areas of the caging environment. We also did not remove

the inoculated ferret prior to the recipient ferret being exposed, because this was already done by Sia and colleagues⁸. In their study it was already demonstrated that SARS-CoV-2 transmission between hamsters via fomites was not as efficient as transmission by direct contact or aerosols/droplets. Since we only collected fur samples and only during the last experiments we were already very careful with our interpretation of the results by stating "In contrast, in the guinea pig study of Asadi et al., up to 650 PFU of infectious influenza virus was recovered from fur of intranasally inoculated animals, which is not a surprise since influenza viruses replicate to much higher titers than SARS-CoV-2 as also shown in Figure 3. Thus, although transmission via aerosolized fomite particles cannot be excluded in the present study, the low amounts of viral RNA and undetectable levels of infectious virus in fur as compared to those in the guinea pig studies makes this a less likely route here". However, if the reviewer or editor desires, we can also leave the part on potential transmission via the fur.

Minor Comments:

1. Lines 115-6, the similar clinical and virological parameters between donor and airborne contact ferrets for all viruses tested is interesting, especially considering the lower dose of virus likely initiating the contact ferret infection. Are the 50% ferret infectious doses known for the viruses studied here, or can this be contextualized to what has been studied in humans?

The ferret infectious dose is not known for the viruses used here. This would require a different experimental set-up involving inoculation of ferrets with decreasing doses of virus. Since the susceptibility of ferrets and humans for infection with the tested respiratory viruses is likely to be different, our results cannot be contextualized to what has been studied in humans.

2. Lines 138-9, the authors conclude that detection of SARS-CoV-2 virus from the fur of directly inoculated ferrets is attributable to grooming. Could this not also be attributed to aerosolized virus released from the inoculated ferret landing on the fur?

We agree with the reviewer and adjusted lines 138-142. However, the observed grooming behavior of ferrets and the high viral load detected in the upper respiratory tract, suggests that grooming is the dominant route of virus transfer to the fur.

3. Authors should specify the temperature and relative humidity at which these experiments were performed in the methods. Authors should also include in the methods section how the modified transmission setup and tubing was decontaminated between experiments.

We added the average temperature and relative humidity to the methods section (lines 479-480) and included a supplementary figure displaying the recordings of each isolator during SARS-CoV and SARS-CoV-2 experiments (Fig S2). For pH1N1 experiments, we did not have dataloggers in place yet. However, as all three experiments were conducted within a narrow time period of two months, we expect a similar temperature and relative humidity during these experiments.

We also added detailed information of our decontamination procedures to the methods section (lines 479-485). The ABSL3+ laboratory contains four class III isolators, each containing one

transmission set-up. As we use four ferret pairs to study the transmissibility of each virus, each set-up is only used once for a particular virus. Therefore, transmission events due to contaminated transmission set-ups can be excluded.

4. Authors mention the “high flow rate” (line 159) that may have carried particles >10um in diameter upward from the inoculated to the donor cage. Did the authors modulate the flow rate (100 L/min) in any experiments to a lower rate to account for this?

We did not modulate the flow rate in any of the experiments. 100 L/min is the standard flow rate used in this kind of experiments, as there is a minimum required fresh air transportation rate to ensure the well-being of animals. In addition, by keeping the flow rate the same during all experiments, we can always compare newly generated transmission data with historical data obtained in the same set-up.

References

1. Linster M, Van Boheemen S, De Graaf M, et al. Identification, characterization, and natural selection of mutations driving airborne transmission of A/H5N1 virus. *Cell*. 2014;157:329–339.
2. Herfst S, Schrauwen EJ, Linster M, et al. Airborne Transmission of INfluenza A/H5N1 Virus Between Ferrets. *Science*. 2009;336:1534–1541.
3. Munster VJ, Wit E De, Brand JM a Van Den, et al. Pathogenesis and Transmission of Swine-Origin 2009 A/H1N1 Influenza Virus in Ferrets. *Science*. 2009;325:481–483.
4. Richard M, Kok A, de Meulder D, et al. SARS-CoV-2 is transmitted via contact and via the air between ferrets. *Nat Commun*. 2020;11:1–6. 2020.
5. Schlottau K, Rissmann M, Graaf A, et al. SARS-CoV-2 in fruit bats, ferrets, pigs, and chickens: an experimental transmission study. *Lancet Microbe*. 2020;1:e218-225.
6. Martina BEE, Haagmans BL, Kuiken T, et al. SARS virus infection of cats and ferrets. *Nature*. 2003;425:915.
7. Asadi S, Gaaloul ben Hnia N, Barre RS, Wexler AS, Ristenpart WD, Bouvier NM. Influenza A virus is transmissible via aerosolized fomites. *Nat Commun*. 2020;11:1–9.
8. Sia SF, Yan LM, Chin AWH, et al. Pathogenesis and transmission of SARS-CoV-2 in golden hamsters. *Nature*. 2020;583:834–838. 2020.

Reviewers' Comments:

Reviewer #1:

Remarks to the Author:

The authors have satisfactorily addressed the concerns raised by this reviewer. Although data on the ferret infectious dose would contribute to the understanding of SARS-CoV-2 transmission in this animal model, the reviewer appreciates that this would require additional experimentation that is beyond the study objectives.

Reviewer #2:

Remarks to the Author:

Authors addressed all comments satisfactorily raised during initial peer review. Inclusion of additional APS data and sequencing data has strengthened the utility and rigor of this manuscript.

Additional comments based on revisions included in the resubmission:

-Table 1, please specify if the days of onset virus shedding in the SARS-CoV direct contact experiment were determined based on virus titers or qRT-PCR.

-Lines 136-7, please include in the methods how fur swabs were collected and stored prior to assessment for viral load, that information appears missing from the text.

-Line 568, please specify how ferret activity was measured and recorded during particle size measurements (visual or video observation? A telemetric device?). And please specify the manufacturer of the aerodynamic particle sizer employed in these studies.

-Line 567 and Figure S1 legend, please specify if this ferret was virus-infected or not, it is currently unclear in the text.

-Figure S1, please specify how many ferrets were evaluated in this capacity as the graph appears to be derived from one animal only. Was there substantial ferret-to-ferret variability of exhaled particle density or size distribution?

Reviewer #1 (Remarks to the Author):

The authors have satisfactorily addressed the concerns raised by this reviewer. Although data on the ferret infectious dose would contribute to the understanding of SARS-CoV-2 transmission in this animal model, the reviewer appreciates that this would require additional experimentation that is beyond the study objectives.

Reviewer #2 (Remarks to the Author):

Authors addressed all comments satisfactorily raised during initial peer review. Inclusion of additional APS data and sequencing data has strengthened the utility and rigor of this manuscript.

Additional comments based on revisions included in the resubmission:

-Table 1, please specify if the days of onset virus shedding in the SARS-CoV direct contact experiment were determined based on virus titers or qRT-PCR.

The information was added as footnote.

-Lines 136-7, please include in the methods how fur swabs were collected and stored prior to assessment for viral load, that information appears missing from the text.

We included a description of how fur swabs were collected in lines 386-388 The storage of the swabs is described in lines 393-394. (Note: we had to change the order of the 'methods' and 'references' sections, so the line numbers in our response differ from those referred to by the reviewers.)

-Line 568, please specify how ferret activity was measured and recorded during particle size measurements (visual or video observation? A telemetric device?). And please specify the manufacturer of the aerodynamic particle sizer employed in these studies.

The activity of the ferret was visually observed and recorded in five minute intervals. This information was added in line 495-496. The manufacturer of the aerodynamic particle sizer was added to lines 490-491.

-Line 567 and Figure S1 legend, please specify if this ferret was virus-infected or not, it is currently unclear in the text.

We specified that an uninfected ferret was used for the measurements.

-Figure S1, please specify how many ferrets were evaluated in this capacity as the graph appears to be derived from one animal only. Was there substantial ferret-to-ferret variability of exhaled particle density or size distribution?

These measurements were produced with one ferret. Hence, we cannot make a statement about a potential ferret-to-ferret variability of exhaled particles. We specified the use of one single animal during these measurements in line 494.